The association between serum lipids and risk of premature mortality in Latin America: a systematic review of population-based prospective cohort studies

Carrillo-Larco Rodrigo M. r.carrillo-larco@imperial.ac.uk rcarrill@ic.ac.uk 1 2 3
Albitres-Flores Leonardo 2 4 5
Barengo Noël C. 6 7 8
Bernabe-Ortiz Antonio 2 9
1 Department of Epidemiology and Biostatistics, School of Public Health, Imperial College London , London , United Kingdom
2 CRONICAS Center of Excellence in Chronic Diseases, Universidad Peruana Cayetano Heredia , Lima , Peru
3 Centro de Estudios de Población, Universidad Católica los Ángeles de Chimbote (ULADECH-Católica) , Chimbote , Peru
4 Facultad de Medicina, Universidad Nacional de Trujillo , Trujillo , Peru
5 Sociedad Científica de Estudiantes de Medicina de la Universidad Nacional de Trujillo-SOCEMUNT , Trujillo , Peru
6 Department of Medical and Population Health Sciences Research, Herbert Wertheim College of Medicine, Florida International University , Miami , United States of America
7 Department of Public Health, Faculty of Medicine, University of Helsinki , Helsinki , Finland
8 Faculty of Medicine, Rı¯ga Stradiņš University (RSU) , Riga , Latvia
9 Universidad Científica del Sur , Lima , Peru
Lau Dennis
Electronic publication date: 2019 Oct 4
Publication date: 2019
Volume: 7
Electronic Location ID: e7856
Received 2019 Jul 1; Accepted 2019 Sep 9
Copyright: ©2019 Carrillo-Larco et al.
Copyright year: 2019
Copyright holder: Carrillo-Larco et al.
License: This is an open access article distributed under the terms of the Creative Commons Attribution License, which permits unrestricted use, distribution, reproduction and adaptation in any medium and for any purpose provided that it is properly attributed. For attribution, the original author(s), title, publication source (PeerJ) and either DOI or URL of the article must be cited.
License URL: https://creativecommons.org/licenses/by/4.0/

Keywords: Dyslipidaemias, Cholesterol, Survival, Latin America

Funding: Strategic Award, Wellcome Trust-Imperial College Centre for Global Health Research 100693/Z/12/Z Imperial College London Wellcome Trust Institutional Strategic Support Fund 294834/Z/16/Z ISSF ICL Wellcome Trust International Training Fellowship 214185/Z/18/Z This work was supported by the Strategic Award, Wellcome Trust-Imperial College Centre for Global Health Research (100693/Z/12/Z), Imperial College London Wellcome Trust Institutional Strategic Support Fund [Global Health Clinical Research Training Fellowship] (294834/Z/16/Z ISSF ICL), and Rodrigo M Carrillo-Larco is supported by a Wellcome Trust International Training Fellowship (214185/Z/18/Z). The funders had no role in study design, data collection and analysis, decision to publish, or preparation of the manuscript.

==============================
Objective

To synthetize the scientific evidence on the association between serum lipids and premature mortality in Latin America (LA).

Methods

Five data bases were searched from inception without language restrictions: Embase, Medline, Global Health, Scopus and LILACS. Population-based studies following random sampling methods were identified. The exposure variable was lipid biomarkers (e.g., total, LDL- or HDL- cholesterol). The outcome was all-cause and cause-specific mortality. The risk of bias was assessed following the Newcastle-Ottawa criteria. Results were summarized qualitatively.

Results

The initial search resulted in 264 abstracts, five (N = 27,903) were included for the synthesis. Three papers reported on the same study from Puerto Rico (baseline in 1965), one was from Brazil (1996) and one from Peru (2007). All reports analysed different exposure variables and used different risk estimates (relative risks, hazard ratios or odds ratios). None of the reviewed reports showed strong association between individual lipid biomarkers and all-cause or cardiovascular mortality.

Conclusion

The available evidence is outdated, inconsistently reported on several lipid biomarker definitions and used different methods to study the long-term mortality risk. These findings strongly support the need to better ascertain the mortality risk associated with lipid biomarkers in LA.

Introduction

An unfavourable serum lipid profile such as increased total cholesterol or LDL-cholesterol is an important determinant of cardiovascular diseases causing large negative health consequences in low- and middle-income countries (LMICs) (GBD 2017 Risk Factor Collaborators, 2018; Global Burden of Metabolic Risk Factors for Chronic Diseases Collaboration, 2014). Understanding the long-term effects of lipids on health is relevant to support the current national and international guidelines which provide recommendations for their management to achieve good cardiovascular health (Catapano et al., 2016; Grundy et al., 2019; NICE, 2014). In addition, many risk scores for primary prevention of cardiovascular diseases include lipid measurements as one of their predictors (Conroy et al., 2003; D’Agostino Sr et al., 2008; Goff Jr et al., 2014; Conroy et al., 2003; D’Agostino Sr et al., 2008). Despite the relevance of serum lipids and use in clinical medicine, the epidemiological research is still limited in LMICs including Latin America (LA) (Ponte-Negretti et al., 2017).

Epidemiological studies have reported inconsistent findings about the association between lipids and premature cardiovascular mortality (Di Angelantonio et al., 2009; Lewington et al., 2007). The Prospective Studies Collaboration reported a lower hazard of death due to ischaemic heart disease for each one mmol/L reduction of total cholesterol (Lewington et al., 2007). However, the evidence of an association between serum lipids and cerebrovascular disease mortality was less consistent in their study (Lewington et al., 2007). Furthermore, the Emerging Risk Factor Collaboration revealed a higher hazard of death due to coronary heart diseases for people with increased cholesterol and LDL-cholesterol levels and those with decreased HDL-cholesterol (Di Angelantonio et al., 2009). Again, the evidence was less conclusive when the main outcome was ischaemic stroke mortality (Di Angelantonio et al., 2009). Other prospective studies have reported that the association between total cholesterol and stroke mortality varies according to stroke sub-type (Yi et al., 2018). Moreover, a study including people aged 60 years and above reported a reduced mortality between increased serum total cholesterol and all-cause mortality, most likely due to a high number of non-cardiovascular deaths (Liang, Vetrano & Qiu, 2017); similarly, LDL-cholesterol seems to have a negative correlation with mortality in people 60 years old and above (Ravnskov et al., 2016). These inconsistent findings have not included populations in LA, where different distribution of cholesterol levels, health profiles and access to healthcare or pharmaceutical treatment (Atun et al., 2015; Cotlear et al., 2015; Farzadfar et al., 2011), exist. Therefore, summarizing studies on the association between lipid biomarkers and mortality in LA populations may complement international evidence, as well as provide valuable information for the development of local guidelines for clinicians and health policy makers. The objective of this study was to synthetize through a systematic review the current scientific evidence on the association between serum lipids and premature mortality in LA.

Methods

Protocol

We conducted a systematic review of the literature following the PRISMA guidelines (Supplemental Information, pp. 02–04) (Liberati et al., 2009). The protocol was registered at PROSPERO (CRD42019120491). Epidemiological studies in adults assessing the association between lipid biomarkers (e.g., total cholesterol) and all-cause as well as cause-specific mortality in LA populations were aimed for. Although no specific comparator was sought, we aimed to study the mortality risk of impaired levels of lipid biomarkers in comparison with recommended levels or per unit change in mg/dL or mmol/L.

Eligibility criteria

We searched for observational prospective cohort studies without any language restrictions regardless of publication time. The study population comprised of individuals from all LA countries. Studies addressing LA people in foreign countries or foreigners in LA countries were excluded. We aimed for population-based studies which had followed a random sampling technique to select the study population. Hospital-based studies, convenient samples or participants selected based on a diagnosis (e.g., patients with diabetes) or risk factor (e.g., obese individuals) were excluded. The exposure of interest was any lipid biomarker, including but not limited to total cholesterol, HDL-cholesterol, LDL-cholesterol or triglycerides.

Information sources

The search was conducted in five data bases: Embase, Global Health and Medline through Ovid, Scopus and LILACS. The search was conducted on December 21st, 2018. No additional sources of scientific information were considered. The search strategy used in these search engines is available in the Supplemental Information, pp. 05–07.

Study selection

The search results were downloaded and compiled in EndNote, where duplicates were identified and excluded. A second search for duplicates was conducted online with Rayyan (Ouzzani et al., 2016). Screening of titles and abstracts was performed by two independent reviewers (RMC-L, LA-F) following the selection criteria above detailed; discrepancies were solved by consensus between them. The full text of the selected reports was studied in detail by the same reviewers, also following the above explained criteria; discrepancies were solved by consensus as well. The two stages of the selection process were conducted with the online tool Rayyan (Ouzzani et al., 2016).

When multiple reports were found for one study, the following algorithm was followed to select one report for inclusion in qualitative synthesis: (i) if they reported on different outcomes (e.g., all-cause and cardiovascular mortality), then all the reports were included; and (ii) the report which analysed the longer follow-up time was included.

Data collection

With the final list of selected reports, information was extracted onto an Excel sheet developed by the authors before data collection started and was not modified afterwards; this form collected similar information as in Ravnskov et al. (2016). The extraction form included information about the study (authors, year of publication, country), about the study population (sample size, age and sex ratio at baseline, follow-up time), and about the distribution of lipid biomarkers including mean and/or prevalence according to data availability in each selected report. Moreover, to assess the mortality risk, risk estimates such as relative risks or hazard ratios were extracted according to what was reported in the original paper. Data extraction was conducted by one reviewer (RMC-L) and independently verified by another one (AB-O); discrepancies were solved by consensus between them.

Risk of bias in individual studies

The Newcastle-Ottawa Scale was used to assess the risk of bias in the selected reports (Wells et al., 2019). The risk of bias assessment was conducted by one reviewer (LA-F) and independently verified by another one (AB-O). If there were discrepancies, these were solved by consensus between these reviewers.

Summary measures

We conducted a qualitative synthesis, and where relevant the risk estimates as described in the original report were summarized. No quantitative synthesis such as a meta-analysis was possible to conduct because of the few retrieved reports, which also exhibited large heterogeneity in the lipid biomarkers assessed, outcomes, and statistical methods.

Ethical considerations

This project was classified as non-human subject research. This is a systematic review of published and open information where no human subjects participated. Approval from an IRB/ethics committee was not necessary.

Results

Study selection

After duplicates were removed, 264 titles and abstracts were screened for eligibility, and 23 were further studied in detail. Three manuscripts were excluded after applying the exclusion criteria: (Cruz-Vidal et al., 1983; Marafon et al., 2003; Sorlie & Garcia-Palmieri, 1990) two studies were excluded because a newer report was available using the same data (Cruz-Vidal et al., 1983; Marafon et al., 2003), and one because the assessed outcome was the same as in another report of the same data (Sorlie & Garcia-Palmieri, 1990). Finally, five reports were included for qualitative synthesis (N = 27,903) (Crespo et al., 2002; Garcia-Palmieri et al., 1981; Garcia-Palmieri et al., 1988; Lazo-Porras et al., 2016; Werle et al., 2011). Of the five selected reports for qualitative synthesis, three were using the same project (Puerto Rico Heart Health Program) (Crespo et al., 2002; Garcia-Palmieri et al., 1981; Garcia-Palmieri et al., 1988) and two were independent studies in Peru (Lazo-Porras et al., 2016) and Brazil (Werle et al., 2011). Figure 1 presents the number of studies at each stage of the selection process and the reasons for exclusion.

Figure 1 Flow-chart of the selection process.

Study characteristics

Three reports collected baseline data in 1965 (Crespo et al., 2002; Garcia-Palmieri et al., 1981; Garcia-Palmieri et al., 1988), one in 1996 (Werle et al., 2011) and one in 2007 (Lazo-Porras et al., 2016). Four reports included middle-aged adults (Crespo et al., 2002; Garcia-Palmieri et al., 1981; Garcia-Palmieri et al., 1988; Lazo-Porras et al., 2016), while one studied the elderly (mean age = 83.6 years) (Werle et al., 2011). Three reports included only men (Crespo et al., 2002; Garcia-Palmieri et al., 1981; Garcia-Palmieri et al., 1988), in one report men accounted for almost half of the study population (47.2%) (Lazo-Porras et al., 2016), and in other report the proportion of men was smaller (36.4%) (Werle et al., 2011). The follow-up time varied from 5 to 12 years (Crespo et al., 2002; Garcia-Palmieri et al., 1981; Garcia-Palmieri et al., 1988; Lazo-Porras et al., 2016; Werle et al., 2011). The outcome of the three reports from Puerto Rico were all-cause mortality (Crespo et al., 2002), cardiovascular disease mortality (Garcia-Palmieri et al., 1988), and cancer mortality (Garcia-Palmieri et al., 1981). Werle et al. (2011) and colleagues studied cardiovascular disease mortality, whereas Lazo-Porras et al. (2016) assessed all-cause and cardiovascular mortality. Details about the study characteristics are shown in Table 1.

Table 1 Characteristics of the selected reports.

Author	Country	Baseline year	Baseline sample size	Baseline age	Baseline % men	Follow-up time	Body mass index (mean)	Smoker	Systolic blood pressure (mean)	Glucose (mean)	Hypertension	Diabetes	
Lazo-Porras et al. (2016)	Peru (Lima, Ayacucho)	2007–08	988	48 (SD: 12)	47	∼5 years	33.3%*	3.3%	NA	NA	NA	NA	
Werle et al. (2011)	Brazil (Veranopolis)	1996	193	84 (SD: 3.3)	36	∼11 years	26.7	6.7%	168	5.4	93%	17.6%	
Garcia-Palmieri et al. (1981)	Puerto Rico	1965	8,793	45–64	100	∼8 years	NA	7.6 ¥	133	5.4	NA	NA	
Crespo et al. (2002)	Puerto Rico	1965	9,136	35–79	100	∼12 years	49.7%*	34.2% ξ	NA	NA	35.4%	NA	
Garcia-Palmieri et al. (1988)	Puerto Rico	1965	8,793	45–64	100	∼12 years	NA	NA	NA	NA	NA	NA	
Notes.

a Overweight prevalence. ¥ number smoked per day. ξ non-smokers.

SD standard deviation

NA not available

For Garcia-Palmieri’s and Crespo’s works the age is given as ranges. Glucose estimates are given as mmol/l.

Lipid biomarkers at baseline

Table 2 shows the means and prevalence estimates for the studied lipid biomarkers across the reports. The selected reports used different classifications to present prevalence estimates. For example, Lazo-Porras et al. (2016) reported the prevalence of low HDL-cholesterol (56.5%), isolated low HDL-cholesterol (36.5%), high non-HDL-cholesterol (91.6%), low HDL-cholesterol with triglycerides ≥200 mg/dL (15.0%), and low HDL-cholesterol with LDL-cholesterol >160 mg/dL (2.0%). Crespo and colleagues showed the prevalence of total cholesterol <200 mg/dL (50.9%), 200–239 mg/dL (33.2%) and ≥240 mg/dL (15.9%) (Crespo et al., 2002).

Table 2 Baseline lipid profile as in the summarised reports.

Author	Baseline Total Cholesterol (mg/dl)	If prevalence, what was the definition?	Baseline HDL(mg/dl)	If prevalence, what was the definition?	Baseline LDL(mg/dl)	If prevalence, what was the definition?	Baseline Triglycerides(mg/dl)	If prevalence, what was the definition?	Baseline Other (specify)	Mean (specify)	If prevalence, what was the definition?	
	Mean (SD)	Prevalence (%)		Mean (SD)	Prevalence (%)		Mean (SD)	Prevalence (%)		Mean (SD)	Prevalence (%)		Mean (SD)	Prevalence (%)			
Lazo-Porras					56.5	Low HDL-cholesterol (HDL-cholesterol <40 in men and <50 in women)		3.0	LDL-cholesterol >160		4.0	Triglycerides ≥200		3.0		LDL-cholesterol >160, triglycerides ≥200 & low HDL-cholesterol	
				36.5	Isolated low HDL-cholesterol		2.0	LDL-cholesterol >160 & triglycerides ≥200								
				91.6	High non-HDL-cholesterol											
				15.0	Low HDL-cholesterol and triglycerides ≥200											
				2.0	Low HDL-cholesterol & LDL-cholesterol >160											
Werle	211.6 (SD: 47.4)			45.5 (SD: 12.6)			139.1 (SD: 42.5)			137.2 (SD: 65.6)			165.4 (SD: 33.7)		ApoA-I (mg/dL)		
												87.5 (SD: 21.0)		ApoB-100 (mg/dL)		
Garcia-Palmieri	202.4									152.0							
Crespo		50.9	<200														
	33.2	200-239														
	15.9	≥240														
Garcia-Palmieri	202.06																
																
																

Other studies reported mean values. For example, Werle et al. (2011) showed that the mean total cholesterol was 211.6 (SD: 47.4) mg/dL, HDL-cholesterol had a mean of 45.5 (SD: 12.6) mg/dL, the mean LDL-cholesterol was 139.1 (SD: 42.5) mg/dL, and the mean for triglycerides was 137.2 (SD: 65.6) mg/dL. Werle et al. also reported on ApoA-I and ApoB-100, with means of 165.4 (SD: 33.7) mg/dL and 87.5 (SD: 21.0) mg/dL, respectively (Werle et al., 2011).

Mortality risk

Table 3 (Supplemental Information p. 08) summarizes the risk estimates as provided by each report. All selected reports analysed different exposure variables not allowing to conduct a meta-analysis. In addition, the reports used different risk estimates including relative risks (RR), hazard ratios (HR) and odds ratios (OR).

Lazo-Porras et al. (2016) studied all-cause mortality based on a composite three-level exposure variable: normal HDL-cholesterol vs isolated low HDL-cholesterol [HDL-c<40 mg/dL in men and <50 mg/ dL in women without hypertriglyceridemia and LDL <160 mg/dL] and non-isolated low HDL-cholesterol [HDL-c<40 mg/dL in men and <50 mg/dL in women accompanied by hypertriglyceridemia and/or LDL ≥ 160 mg/dL]; the first level was the reference category whereas the second and third exhibited a RR of 1.11 (95% CI [0.49–2.51]) and 0.82 (95% CI [0.21–3.15]), respectively (Lazo-Porras et al., 2016). Crespo and colleagues also studied all-cause mortality in a cohort of men, showing 3% higher mortality risk in people with high total cholesterol in comparison to their peers with total cholesterol in the normal range; in addition, there was not higher risk when the latter group was compared with people with borderline high total cholesterol (Crespo et al., 2002).

Table 3 Risk estimates of lipid biomarkers on mortality as in the summarised reports.

Garcia-Palmieri et al. (1981) reported multiple regression coefficients (“The logistic function was used to model the relationship between cancer mortality and serum cholesterol”), so that the OR herein reported corresponds to the exponential function of the coefficients in the manuscript. Crespo et al. (2002) - The authors did not provide risk estimates; the IRR herein reported corresponds to the relation between crude death rates among people with high cholesterol (crude death rate = 16.3%), borderline high blood cholesterol (crude death rate = 15.5%) and desirable cholesterol (crude death rate = 15.8%). For example, 15.8/16.3 ∼0.97. Where available, the risk estimates herein summarized correspond to the adjusted models reported in the original reports.

Author	Exposure definition	All-cause mortality	Cardiovascular mortality	Other causes of death assessed	
		Risk estimate	lower IC	upper IC	Risk estimate	lower IC	upper IC	Risk estimate	lower IC	upper IC	Cause?	
Lazo-Porras et al. (2016)	Normal HDL vs isolated low HDL [HDL-c <40 mg/dL in men and <50 mg/ dL in women without hypertriglyceridemia and LDL <160 mg/dL]	RR = 1.11	0.49	2.51	RR = 0.45	0.05	4.24					
Normal HDL vs non-isolated low HDL [HDL-c <40 mg/dL in men and <50 mg/dL in women with hypertriglyceridemia and/or LDL ≥160 mg/dL]	RR = 0.82	0.21	3.15	RR = 3.67	0.48	28.16					
Werle et al. (2011)	LDL (mg/dL)				HR = 1.00	0.99	1.01					
HDL (mg/dL)				HR = 1.01	0.97	1.05					
ApoA-I (mg/dL)				HR = 0.99	0.98	1.01					
Garcia-Palmieri et al. (1981)	Serum cholesterol (mg/dL), 45–54 rural men							OR = 0.54	p < 0.05		Cancer	
Serum cholesterol (mg/dL), 55–64 rural men							OR = 0.48	p < 0.05		Cancer	
Serum cholesterol (mg/dL), 45–54 urban men							OR = 0.78			Cancer	
Serum cholesterol (mg/dL), 55–64 urban men							OR = 1.08			Cancer	
Crespo et al. (2002)	Desirable total cholesterol ( <200 mg/dL) vs high total cholesterol (≥240 mg/dL)	IRR = 0.97										
Desirable total cholesterol ( <200 mg/dL) vs borderline high blood cholesterol (200-239 mg/dL)	IRR = 1.02										
Garcia-Palmieri et al. (1988)	Serum cholesterol (mg/dL)				OR = 0.004	p < 0.05						

Werle and colleagues studied cardiovascular mortality associated with one-unit change (mg/dL) in LDL-cholesterol (HR = 1.00, 95% CI [0.99–1.01]), HDL-cholesterol (HR = 1.01, 95% CI [0.97–1.05]), and Apo-A (HR = 0.99, 95% CI [0.98–1.01]) (Werle et al., 2011). Garcia-Palmieri’s team reported on cardiovascular mortality too, though the magnitude of the effect was very small (Garcia-Palmieri et al., 1988).

Garcia-Palmieri et al. (1988) used cancer mortality as main outcome reporting the OR for one-unit change (mg/dL) in total cholesterol by urban/rural location and age group. For example, in rural men aged 45–54 years, the OR was 0.54 (p < 0.05) for each mg/dL increase in total cholesterol (Garcia-Palmieri et al., 1981).

Risk of bias

Table 4 presents the summary of the risk of bias assessment, details in Supplemental Information p. 09. (Lazo-Porras et al., 2016) authored the work with the least risk of bias, whereas both reports by Garcia-Palmieri (Garcia-Palmieri et al., 1981; Garcia-Palmieri et al., 1988) showed the highest risk of bias mostly due to the comparability criteria.

Table 4 Risk of bias assessment.

	selection	Comparability	Outcome	
Lazo-Porras et al. (2016)	★★★★	★★	★★★	
Werle et al. (2011)	★★★	★★	★★★	
Garcia-Palmieri et al. (1981)	★★★	★	★★★	
Crespo et al. (2002)	★★★★	★★	★★	
Garcia-Palmieri et al. (1988)	★★★	★	★★★	

A narrative experience

Lazo-Porras et al. (2016) and colleagues studied the residual dyslipidaemic profile and its impact on mortality. Interestingly, they analysed a cohort of rural dwellers, urban people and rural-to-urban migrants (Lazo-Porras et al., 2016). These populations have unique features and these results could still be of interest to other LMICs where internal migration and urbanization is underway. However, one could argue on the pragmatic need, implications and applicability of these results in health policy or clinical practice.

Werle et al. (2011) reported on elderly individuals with relatively strong risk estimates. As aging is an ongoing process in LA and LMICs, these results and their implications among the oldest old could inform future studies in these populations. Nonetheless, this endeavour could not provide further arguments to advance the inconsistent findings among elderly populations, as signalled in the introduction.

Garcia-Palmieri et al. and Crespo et al. analysed a cohort of men starting in 1965, and studied all-cause, cardiovascular disease and cancer mortality (Crespo et al., 2002; Garcia-Palmieri et al., 1981; Garcia-Palmieri et al., 1988). This is the largest cohort herein summarized where outcomes were comprehensively adjudicated; however, because only men were included, these estimates could not successfully inform clinical practice or public health for the whole population. Although these authors analysed cardiovascular mortality, and so did Lazo-Porra’s and Werle’s team (Lazo-Porras et al., 2016; Werle et al., 2011), they did not look at specific cardiovascular events, e.g., coronary heart disease, ischaemic stroke or haemorrhagic stroke. Therefore, these reports could not provide additional evidence to elucidate the inconclusive knowledge signalled in the introduction.

Overall, total cholesterol was the exposure mostly studied, except for Werle’s and Lazo-Porras’s work, which included LDL-cholesterol and residual dyslipidaemic profile, respectively (Lazo-Porras et al., 2016; Werle et al., 2011). The effect of total cholesterol on health outcomes such as mortality, could vary depending on its composition, i.e., whether LDL-cholesterol levels are high. Conversely, LDL-cholesterol is a well-stablished cardiovascular risk factor for which successful pharmacological treatment is available (e.g., statins). Unfortunately, prospective evidence on the effect of LDL-cholesterol is still scarce in LA, though much needed to inform clinical practice as well as resources and treatment allocation.

Discussion

Summary of evidence

This systematic review of the literature in LA did not reveal scientific evidence on an association between unfavourable serum lipid biomarkers and premature mortality in the general population. Furthermore, the definitions used to categorize lipid biomarkers were inconsistent across reports. In addition, only one study was conducted within the last ten years. Overall, our findings call to either conduct new cohort studies or use available ones to systematically estimate the mortality risk associated with lipid profiles, using consistent metrics and clinically relevant definitions. Thus, there is a need to study the long-term effects of lipid profiles as this will provide evidence to inform local clinical practice, health policy and priority setting for LA.

Limitations of the review

Even though we conducted a comprehensive literature search, including a LA-based search engine (LILACS), we did not systematically search grey literature such as conference abstracts. We argue that, even if these sources had provided additional references, these would contain probably limited information.

Although some of the selected studies reported on relevant lipid biomarkers such as LDL-cholesterol, they failed to analyse clinically relevant definitions. For example, Lazo-Porras and colleagues reported on several combinations of lipid metrics (e.g., low HDL-cholesterol and triglycerides ≥200 mg/dL), but did not report on high LDL-cholesterol, which happens to be the lipid mainly targeted by pharmacological treatment for cardiovascular prevention (Collins et al., 2016; Grundy et al., 2019; Yusuf et al., 2016).

Results in context

Previous large individual-level meta-analysis have assessed mortality risk associated with one unit change in lipids (Di Angelantonio et al., 2009; Lewington et al., 2007). In this systematic review, the most recent study addressing this exposure was the work by Werle et al., whom reported that there was no strong evidence of higher risk (Werle et al., 2011). Noteworthy, the study population in Werle’s work had a mean age of 83 years (Werle et al., 2011). Therefore, this finding is consistent with previous reports where the magnitude of the risk estimates would also decrease with age (Lewington et al., 2007). Other recent systematic review studying people aged 60 years and above, also reported an inverse association between LDL-cholesterol and all-cause mortality (Ravnskov et al., 2016). Despite this evidence, the use of statins in elderly still seems to reduce cardiovascular mortality (Cholesterol Treatment Trialists’ Collaboration, 2019). Evidence in LA regarding young adults, middle-aged adults and elderly, is still limited to draw conclusions and formulate strong recommendations. For the time being, international guidelines should be followed along with clinical reasoning and shared decision making.

The INTERHEART, a case-control global endeavour studying myocardial infarction, reported that the association between this cardiovascular outcome and total cholesterol as well as non-HDL cholesterol was small in LA, in comparison to other world regions (McQueen et al., 2008). Although this is a large and relevant scientific contribution, the results have the limitations of any case-control study. In addition, this was conducted almost ten years ago. Their findings deserve further verification with a stronger study design and more recent observations, to provide robust evidence that can be introduced in clinical and public health practice and that can account for the current trends in cardio-metabolic risk factors.

Relevance for LA

Strong scientific evidence is needed to develop successful policies, inform resources allocation, and advance clinical practice. Regarding lipid biomarkers and its associated mortality risk, much research is needed. A proposed call to action for LA is presented in Fig. 2.

Figure 2 Call to action for Latin America in research and policy.

Despite limited scientific research in the field of lipid biomarkers in LA in general, there is a growing interest about this health profile in the clinical and public health communities of LA (Ponte-Negretti et al., 2017). This interest has led to the development of local guidelines on management of hyperlipidaemias in some countries such as Mexico (Secretaria de Salud, 2013) and Colombia (Sistema General de Seguridad Social en Salud Colombia, 2014; Sistema General de Seguridad Social en Salud Colombia, 2014) among others (Caja Costarricense de Seguridad Social, 2004; Sociedad Argentina de Cardiologia, 2018; Sociedad Uruguaya de Aterosclerosis, 1998; (Caja Costarricense de Seguridad Social, 2004; Sociedad Argentina de Cardiologia, 2018; Sociedad Uruguaya de Aterosclerosis, 1998). Analysing whether these local guidelines are in accordance with international recommendations or current scientific evidence is beyond the scope of this work, but definitely merits a close inspection.

To the best of our knowledge, no relevant international organization has set goals, targets or health and research policies for the management and control of lipid biomarkers or dyslipidaemias in LA. Although a group of practitioners and researchers of LA has published a regional consensus highlighting key features of lipid profiles in LA (Ponte-Negretti et al., 2017), further scientific evidence is needed for this momentum to foster research, policy and clinical practice. In addition to a consensus, additional pragmatic steps are needed (Ponte-Negretti et al., 2017). For example, we recommend that this group (Ponte-Negretti et al., 2017) or other relevant professional or governmental body, issues a list of basic metrics that should be included and reported in any research studying lipid biomarkers in LA. These measures may allow to conduct meta-analysis and to estimate other population health metrics benefiting of consistent and comparable lipid-related metrics throughout LA.

Most of the risk estimates used in the studies was total cholesterol. This lipid biomarker, although relevant and inexpensive, does not allow to identify whether a reduction of LDL-cholesterol or an increase of HDL-cholesterol is needed. A key determinant of lipid fraction is diet. Certain foods will increase LDL-cholesterol whilst others will improve HDL-cholesterol and viceversa (Forouhi et al., 2018; Schwingshackl et al., 2018). Because LA shows great variability in diet patterns between and within countries, this could define higher/lower levels of different lipid fractions. Future studies should try to ascertain lipid fractions in addition to total cholesterol.

Other determinants of lipid profiles are weight status and physical activity, whereby obesity increases LDL-cholesterol and decreases HDL-cholesterol whilst physical activity reduces LDL-cholesterol. Overweight and obesity have dramatically increased across LA in the last decades (NCD Risk Factor Collaboration, 2017). Also, a global analysis reported that women in LA have one of the largest prevalence estimates of physical inactivity (Guthold et al., 2018). Although these risk factors need to be addressed on their own, these alarming trends also call to improve the study of lipid biomarkers in LA at the general population level. In this line, government and international agencies could potentiate national surveys to also collect lipid biomarkers such as in Mexico (Aguilar-Salinas et al., 2010), Ecuador (Freire et al., 2014) and Chile (Margozzini & Passi, 2018). If in the next years these could be linked to death registries so that individual risks could be estimated, a major step forward in the study of lipid biomarkers would be achieved.

Conclusions

To date, it is not possible to ascertain the association between lipid biomarkers and mortality risk in LA. The available evidence is outdated, and the definitions of lipid biomarkers are inconsistent. In addition, different methods were used to measure the long-term mortality risk in LA populations. These findings strongly suggest conducting larger studies within the LA population to get valuable risk estimates of the associations between serum lipids and premature mortality.

Supplemental Information

Supplemental Information 1 PRISMA checklist

Click here for additional data file.

Supplemental Information 2 Rationale and contribution of this work

Click here for additional data file.

Supplemental Information 3 Supplementary material including search terms and risk of bias assessment

Click here for additional data file.

Additional Information and Declarations

Competing Interests

Author Contributions

Data Availability

The authors declare there are no competing interests.

Rodrigo M. Carrillo-Larco conceived and designed the experiments, performed the experiments, analyzed the data, prepared figures and/or tables, authored or reviewed drafts of the paper, approved the final draft.

Leonardo Albitres-Flores performed the experiments, analyzed the data, prepared figures and/or tables, authored or reviewed drafts of the paper, approved the final draft.

Noël C. Barengo analyzed the data, authored or reviewed drafts of the paper, approved the final draft.

Antonio Bernabe-Ortiz conceived and designed the experiments, analyzed the data, authored or reviewed drafts of the paper, approved the final draft.

The following information was supplied regarding data availability:

This is a systematic review of the literature. No raw data was analysed.

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
