# Peer review of "The association between serum lipids and risk of premature mortality in Latin America: a systematic review of population-based prospective cohort studies"

_PeerJ, doi:10.7717/peerj.7856_

## Round 0.1 · original submission · Major Revisions

Your work has received two independent peer reviews. However, it can be further improved to improve readability for our readers. Please respond in particular to the comments from reviewer 2.

I suggest significant re-structuring to address the different facets of cholesterol and mortality from the studies identified from your systematic review.

As the studies are so diverse, combining them into table form akin to usual systematic review does not make good reading. (Table 2).

As the populations in the included studies are quite diverse, it would be beneficial to include more population/demographic/risk factor characteristics in table format.

In discussing mortality, you have limited to 3 of the included studies only. Perhaps with the re-structuring, you can highlight the individual study better.

A table form (bullet points) for a 'call to action for latin america' may also improve readability.

Reviewer 1 ·

Basic reporting

Thank you for the opportunity to review this paper. It is well written with clear English throughout.

Experimental design

This is a well conceived and designed study. I particularly congratulate the authors on refraining from performing quantitative analysis on such heterogeneous studies, but instead opt to rigorously appraise the literature and identify a clear need for specific studies to look at the Latin America region.

Validity of the findings

The unique aspect of this systematic review is its population. In this regard, the findings are novel in that there are not enough comparable studies that can be assessed quantitatively. There exists a need for further studies to establish whether or not a link exists between lipid biomarkers and mortality in Latin America.

Additional comments

Thank you for a well-conducted and a well-written study.

Reviewer 2 ·

Basic reporting

The overall reporting structure is generally good.

There are some phrasing/grammatical issues which require attention:
e.g. line 40 (Abstract) "To synthesise the scientific evidence ON OF the association between serum LIPID and premature mortality...". Final English proof reading recommended to tidy up a number of minor issues.

Introduction:
- In describing the "inconsistent findings" of epidemiological data, the authors only really highlight two issues that appear inconsistent:
i. The weaker asssociation between unfavourable lipid profiles and stroke/mortality
ii. The weaker association between unfavourable lipid profiles and older age
- These are not really issues that define the purpose of the manuscript, which aims to examine the association betweeen "serum lipid and premature mortality in Latin America

Experimental design

The methods are well described, with a robust, systematic approach to data retrieval and collection.

Validity of the findings

The major problem with this manuscript is that the results of the literature search yielded only three individual population studies, all of which examined and reported on very different questions.

Crespo et al aimed to examine the relationship between physical activity and obesity with all-cause mortality in Puerto Rican Men, and reported some outcomes according to varied levels of total cholesterol.

Werle et al examined risk factors for the very elderly in a city in Brazil.

Lazo-Porras et al examined the residual dyslipidaemic profile and its impact on mortality.

As the authors have correctly identified, this makes a meta-analysis of the findings impossible. As such, the manuscript "qualitatively" describes the findings of these three studies.

Unfortunately for the researchers, in attempting to uniformly collect and present the data, it makes reading the results more difficult.

I think in such a setting, where the literature search has identified three largely different studies, where meta-analysis is impossible, the authors would be better off writing a narrative review, where discussions regarding issues relating mortality and :
- TC, LDL-C
- residual dyslipidaemic profiles
- elderly lipid profiles
- lipids and coronary vs stroke outcomes

can all be separately addressed and relevant literature can be presented in each section, combined with relevant literature from Western/other populations, to contextualise it all.

Additional comments

I congratulate the methodical approach to this paper and appreciate the hard work involved, but I feel presenting your findings as is makes it overall confusing to follow, since the three papers analysed produced three very different sets of results.

---

## Round 0.2 · Minor Revisions

Thank you for the improvements made to the manuscript.
Please consider making the following remaining minor changes:

Table 1: Age (Please avoid using 2 decimal points - suggest rounding and add standard deviation or median/range); Percentange men (please avoid decimal points); Smoker (maximum 1 decimal point); systolic BP (avoid decimal point); Glucose (please state international units); for blank spots, use NA (not available).

Table 2: Prevalence (use %). Streamline all I.U. for lipids to mg/DL in the table header

Reviewer 1 ·

Basic reporting

Thank you for reviewing the manuscript which now reads better and conveys the message in a clearer way.

Experimental design

No concerns. The paper is limited by the number of relevant studies identified but this is acknowledged and discussed adequately by the authors.

Validity of the findings

No concerns.

Additional comments

Thank you for the opportunity to review this paper. I am satisfied with the revised form and have no further concerns.

---

## Round 0.3 · accepted · Accept

Thank you for improving this work to publication standard.